# Household Survey of Trachoma among Children Living in Pernambuco, Brazil

**DOI:** 10.3390/pathogens8040263

**Published:** 2019-11-25

**Authors:** Cintia Michele Gondim de Brito, Celivane Cavalcanti Barbosa, Sérgio Murilo Coelho de Andrade, André Luiz Sá de Oliveira, Ulisses Ramos Montarroyos, Cristiano Ferraz, Marcel de Toledo Vieira, Maria de Fátima Costa Lopes, Giselle Campozana Gouveia, Zulma Maria de Medeiros

**Affiliations:** 1University of Pernambuco, Recife 50100130, Pernambuco, Brazil; ulisses.montarroyos@upe.br (U.R.M.);; 2Department of Epidemiological Surveillance, First Healthcare Region of the Health Department of the State of Pernambuco, Recife 50050911, Pernambuco, Brazil; celivane.cb@gmail.com; 3Aggeu Magalhães Institute (IAM), Oswaldo Cruz Foundation (FIOCRUZ), Recife 50740465, Pernambuco, Brazil; sergiomcoelho@gmail.com (S.M.C.d.A.); andre.sa@cpqam.fiocruz.br (A.L.S.d.O.); firegisa@hotmail.com (G.C.G.); 4Federal University of Pernambuco, Recife 50670901 Pernambuco, Brazil; cferraz.stat@gmail.com; 5Federal University of Juiz de Fora, Minas Gerais 36036900, Brazil; mdtvieira@gmail.com; 6Ministry of Health, Brasília 70058900, DF, Brazil; mariaf.lopes@saude.gov.br

**Keywords:** neglected diseases, trachoma, *Chlamydia trachomatis*, epidemiological surveys, health inequalities

## Abstract

This study analyzed the association between individual and household factors and the incidence of trachoma among a population aged between 1 and 9 years in the state of Pernambuco. This was a population-based household study conducted using a population-based sample of residents from 96 census sectors of the 1778 sectors considered to be at social risk in the state. The estimated odds ratio of the univariate analysis presented a confidence interval of 95%. Weights and clusters were adjusted through the Generalized Linear and Latent Mixed Model (GLLAM) method. Trachoma cases were the dependent variable in the multivariate analysis. The independent variables were selected through the stepwise forward method, with an input criterion of 20% (p < 0.20) and an output criterion of 10% (p < 0.10). The prevalence was 6.65%. Trachoma was associated with a female sex, age of 5–9 years, either the absence of use or infrequent use of soap to wash the hands and face, the presence of nasal secretion, a lack of piped water from a public supply system, a greater number of rooms used for sleeping, a greater number of people living in the same household, and a family income of up to one minimum monthly wage. The prevalence of follicular trachoma in Pernambuco was higher than what is recommended by the World Health Organization (WHO).

## 1. Introduction

Associations of socioenvironmental factors within the field of healthcare have gained attention from the scientific community and international organizations [1]. Areas with inadequate sewage collection and treatment, precarious access to healthcare services, poor housing conditions, and low educational levels favor occurrences of neglected diseases [2], such as trachoma [3].

Trachoma is still considered to be the most important cause of avoidable blindness in the world [4]. The only source of transmission is humans with active trachoma [5]. *Chlamydia trachomatis* can be transmitted either from person to person or indirectly, by means of sharing contaminated objects, in addition to mechanical transmission through vectors, such as insects like the housefly [6]. The World Health Organization (WHO) has stated that the endemic disease will have been brought under control when the prevalence of active trachoma is <5% [7].

Poor and rural areas of 41 countries in Africa, Central and South America, Asia, Australia, and the Middle East are hyperendemic [7]. Since 2002, studies among schoolchildren have revealed a trachoma prevalence of >5% in several Brazilian states [8,9,10].

To reach the global elimination of trachoma as a cause of blindness by 2020, WHO established a goal of carrying out population-based surveys among samples to verify the epidemiological situation [7,11,12] and implementing the SAFE strategy to control trachoma (S—surgery in cases of trachomatous trichiasis; A—antibiotic therapy in cases of active trachoma; F—facial hygiene; and E—environmental improvements) [13]. Use of the SAFE strategy has been shown to be a tool for reducing the prevalence of active forms of the disease [14].

In Brazil, which is a signatory to resolution 51/2011 [15], pilot surveys have been conducted among populations that are considered to present a social risk in two states: Pernambuco and Tocantins. Based on a household survey conducted in Pernambuco, the objective of the present study was to ascertain individual and household risk factors for trachoma among children aged 1 to 9 years.

## 2. Materials and Methods 

### 2.1. Study Area

The present study was conducted in the state of Pernambuco, which is located in the northeastern region of Brazil. This state covers an area of 98,076,001 km^2^ and is divided into 184 municipalities, distributed into five mesoregions: Metropolitan Region of Recife, Zona da Mata, Agreste, Sertão do São Francisco, and Sertão Pernambucano (Figure 1) [16]. The estimated population in 2014 was 9,277,727 inhabitants [17].

### 2.2. Study Design

This was a population-based study completed using data from the national household survey on trachoma that was conducted between 2014 and 2015.

### 2.3. Population Characterization

The sample selection parameters were the prevalence of active trachoma of 5%, confidence level of 95%, maximum margin of error of 0.02, and correction factor for a finite population with an effect of 4 [18]. The target population comprised 1778 census sectors that met the following social risk criteria: *at least 50% of households with per capita income of up to ¼ of the minimum monthly wage and percentage of households connected to the general water supply network below 95%* [17]. Of the 1778 eligible sectors, a randomized sample of 96 census sectors in which children between the ages of 1 and 9 years were living was selected. All residents were examined.

### 2.4. Sample Collection

A questionnaire was applied in the eligible households. It comprised individual questions (on sex, age group, use of bath and face towels, use of soap to wash the face and hands, whether the person slept alone, and presence of nasal secretion) and household questions (on type of home, water from the public network, any intermittence of water supply, type of sewage system, destination of sewage, destination of solid waste, flies in the household, number of rooms used for sleeping, educational level of the head of the family, family income, and number of people living in the household).

Cases of trachoma were diagnosed through an external eye examination, using a magnifying glass (2.5×) and either natural or artificial light, and were conducted by trained and standardized examiners [5].

The cases of trachoma among children and members of their families were classified as described by Thylefors et al. (1987) [19]. All cases were treated in accordance with the recommendations of the Ministry of Health [5].

### 2.5. Data Management and Analysis

The data were analyzed using STATA (version 12), with the exclusion of missing values. The data were adjusted by applying a correction factor [20] to account for random effects and cluster sample sizes [21,22].

The frequencies of all variables relating to the state of Pernambuco and its five mesoregions were defined and an association analysis of independent variables in relation to the dependent variable was then conducted through univariate analysis. Odds ratios were estimated with a 95% confidence interval. Weights and clusters were adjusted through the Generalized Linear and Latent Mixed Model (GLLAM) method [22].

A multivariate analysis was conducted using trachoma cases as the dependent variable. The independent variables were selected through the stepwise forward method, with an input criterion of 20% (p < 0.2) and an output criterion of 10% (p < 0.10).

An agglomeration indicator (AI) was built based on the following formula:AI=xy
where

*x*: No. of people residing in the household;

*y*: Mean no. of rooms used for sleeping = No. of rooms/No. of households.

### 2.6. Ethics Statement

This research project was approved by the Ethics Committee of the Aggeu Magalhães Institute, Fiocruz, Pernambuco (CAEE 21192013.0.0000.5190).

## 3. Results

A total of 4238 households were evaluated, which presented 446 cases of trachoma among the 7423 children examined. The prevalence of trachoma in the state was 6.65% (CI 5.39–8.17).

The variables for the final logistic regression model (in bold) were selected in the univariate analysis, presented in Table 1 and Table 2: five individual variables (sex, age group, use of a bath and face towel, use of soap to wash the face and hands, and presence of nasal secretion) and eight household variables (water from the public network, intermittence of water supply, type of sewage, destination of sewage, destination of solid waste, flies in the household, number of rooms used for sleeping, and number of people living in the household).

Table 3 presents the final adjusted model, in which trachoma is associated with the following individual variables: being a girl between the ages of 5 and 9 years and not using or only sometimes using soap to wash the face and hands. In the state of Pernambuco, children who presented nasal secretion had a 94% higher chance of having trachoma.

The following household characteristics were associated with trachoma: not having a piped water supply from the public network, a greater number of rooms used for sleeping, and a greater number of people in the household. These factors increased the chances of occurrence of the disease (Table 3). Most homes had two rooms for sleeping, with an agglomeration indicator of 2.2 people per room.

## 4. Discussion

An analysis on the risk factors for trachoma provides important information for planning and implementing actions in disease control programs [5,14,15,23]. The present study revealed factors that were associated with occurrences of the disease, using households as the database.

Among the factors investigated, it was observed that girls had greater chances of contracting trachoma in Pernambuco and in the mesoregion of Zona da Mata. This association has also been reported in population-based surveys in Senegal [24], Ethiopia [25], Gambia, and Tanzania [13]. This is probably because girls more frequently help in tending to younger siblings and because affective behavior is closer among girls [13,25]. However, another explanation could be a higher susceptibility to infection by *C. trachomatis* among females, as reported by Ngondi et al. (2008) [25].

In the present population-based study, as in studies conducted in Africa [18,26,27], children of school and preschool ages were assessed. However, unlike studies from the African continent, the results from the present investigation demonstrated that children of a school age had greater chances of presenting the disease than preschool children. This is possibly explained by reinfections that occur with an increasing age [18].

Studies conducted in Brazil have shown that school-age children (5–9 years) have greater chances of presenting trachoma [10,28,29], thus corroborating the results of the present study. Last et al. (2014) [18] showed that cases occurred predominantly among children between 0 and 5 years of age, in a household survey conducted in Guinea-Bissau. This was similar to what was observed in a school survey conducted in Brazil between 2002 and 2008 [8,9]. Schools are frequently the environment for conducting evaluations and interventions in relation to diseases because children are easily accessible and available in this setting. In the specific case of Brazil, school surveys are unable to characterize trachoma among children between the ages of 1 and 4 years [30], because most individuals of this age group are not enrolled in public schools.

Facial cleaning is among the recommendations of the SAFE strategy [31,32,33,34,35,36]. A greater risk of disease transmission persists when hygiene behaviors are not translated into routine attitudes [37]. The prevalence of trachoma is reduced through a higher frequency of facial cleaning [10,29,35,38].

The relationship between a clean face and reduced chances of trachoma is one of the strongest associations found in the literature [29,38,39,40]. In Pernambuco, a lack of the use or only occasional use of soap to wash the hands and face increases the risk of transmitting trachoma by more than 60%, in comparison with the regular use of soap. In some mesoregions of this state, this risk was tripled or even higher. In Brazil, the school health program, a partnership between healthcare and educational bodies, can conduct combined actions to direct children to clean their faces and hands as a way to decrease transmission of this disease, since most cases have been observed among school-age children.

*C. trachomatis* can be found in nasal secretion [18,41], which can increase the chance of transmission among children living in endemic areas. The presence of nasal secretion among children in Pernambuco demonstrates that this factor was associated with occurrences of trachoma.

Among the household factors, a lack of water supply from the public network in Pernambuco was shown to increase the chances of having trachoma. In this state, intermittence of the water supply has led to people storing water in barrels, buckets, or water tanks to ensure that they have a supply for domestic consumption. When there is no piped public water network, water is drawn from wells, mines, or *cacimbas*. The results found may be related to the amount and/or quality of water to which the population has access, since intermittence of the water supply leads to an inadequate storage or use of water of an uncontrolled quality [27,38,42,43,44]. Moreover, with little water available, hygiene actions are less frequent. Studies in Gambia [45] and Ethiopia [46] demonstrated that there was a reduction in the transmission of *C. trachomatis* infection with improved sanitation and water access.

In Pernambuco and in the mesoregions of the Metropolitan Region of Recife and Sertão do São Francisco, the more rooms that there were in a household, the greater the risk of trachoma was. This result goes against the results found in other surveys conducted in Brazil, which indicated that a smaller number of rooms presented a higher risk of trachoma [10,29].

An agglomeration indicator was built with the aim of understanding this phenomenon. It was found that larger households concentrated more residents, which increased the chance of transmission, even though there were more rooms that could be used for sleeping. According to Favacho et al. (2018) [29], the greater the number of people in a household was, the higher the risk of the disease was. Hence, larger families contributed more towards the incidence of infection [13]. Moreover, with more people, there was a greater chance of sharing beds. In the mesoregions of Agreste and Sertão Pernambucano, most cases identified were from those who shared their bed with other residents. This could explain the maintenance of the transmission cycle of the disease within households with greater numbers of people that was found in the present study.

The target population of the present investigation was selected using social risk criteria. Therefore, it was expected that there would be no significant differences between individuals earning up to one minimum monthly wage and more than one minimum monthly wage. In the final model, it was seen that in the mesoregion of Sertão Pernambucano, people earning up to one minimum monthly wage were more likely to contract the disease. This association, in addition to the social criteria established, related to a population that resided in an area of the state with a low development index [47].

Population-based surveys imply operational difficulties and high costs. Trachoma is a disease that is known to be related to poverty [3]. Therefore, the team that conducted the national survey opted to work with the social risk criteria, which may have led to a limitation in the present study since this may represent a bias in sample selection. These options were implemented to fill the gap of knowledge about this disease in silent areas that nevertheless form part of populations living in areas of extreme poverty.

## 5. Conclusions

The prevalence of follicular trachoma in Pernambuco was higher than what is recommended by the WHO. The factors that were associated with maintaining the transmission chain of trachoma were girls of a school age, those who did not have the habit or had an infrequent habit of cleaning their hands and face, individuals who did not have access to water from the public network, individuals who were part of larger families, and individuals whose family income was up to one minimum monthly wage.

The present study demonstrates the need for monitoring and surveillance measures aimed towards trachoma. These need to be implemented together with intersectoral actions to promote health and improve socioeconomic, environmental, and educational conditions, in order to reduce the transmission of trachoma.

## Figures and Tables

**Figure 1 pathogens-08-00263-f001:**
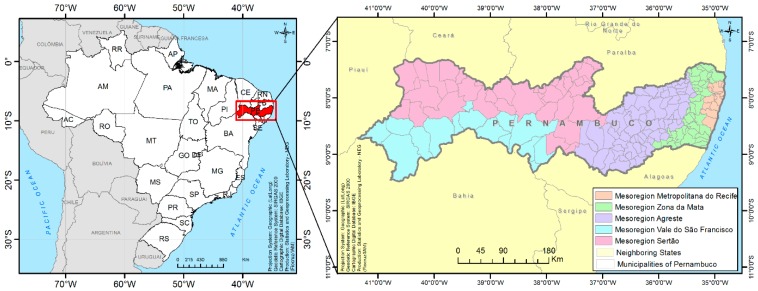
Geographical location of the state of Pernambuco and its division into mesoregions.

**Table 1 pathogens-08-00263-t001:** Univariate analysis on variables relating to individual characteristics of cases of trachoma in the population between 1 and 9 years of age investigated in Pernambuco and mesoregions, 2014–2015.

Individual Characteristics	Pernambuco			Metropolitan Region of Recife			Zona da Mata			Agreste			Sertão do São Francisco			Sertão Pernambucano		
	N	+	% ^a,b^	N	+	% ^a,b^	n	+	% ^a,b^	n	+	% ^a,b^	n	+	% ^a,b^	n	+	% ^a,b^
**Sex**																		
Male *	3849	208	5.40	793	45	5.67	690	26	3.77	834	68	8.15	691	43	6.22	841	26	3.09
Female	3574	238	6.66	742	56	7.55	632	37	5.85	821	80	9.74	670	34	5.07	709	31	4.37
OR ^c^ (95% CI)	1.38 (1.06–1.80)			1.39 (0.84–2.32)			1.62 (0.95–2.77)			1.39 (0.83–2.31)			0.88 (0.33–2.31)			1.42 (0.80–2.49)		
p-value	**0.016**			**0.196**			**0.074**			0.208			0.798			0.223		
**Age Group**																		
From 1 to 4 *	2922	155	5.30	633	37	5.85	523	28	5.35	601	47	7.82	570	25	4.39	595	18	3.03
From 5 to 9	4501	291	6.47	902	64	7.10	799	35	4.38	1054	101	9.58	791	52	6.57	955	39	4.08
OR ^c^ (95% CI)	1.29 (0.98–1.68)			1.28 (0.77–2.12)			0.79 (0.46–1.36)			1.46 (0.87–2.42)			0.78 (0.27–2.19)			1.35 (0.88–2.07)		
p-value	**0.061**			0.338			0.400			**0.149**			0.638			**0.171**		
**Use of Bath Towel**																		
Yes *	7391	444	6.01	1528	100	6.54	1315	63	4.79	1643	148	9.01	1356	76	5.60	1549	57	3.68
No	26	2	7.69	3	1	33.33	6	0	0	11	0	0	5	1	20.00	1	0	0
OR ^c^ (95% CI)	2.13 (0.19–23.63)			21.33 (0.25–1787.22)			Not calculated			Not calculated			339.52 (0.62–184779.60)			Not calculated		
p-value	0.538			**0.176**			-			-			**0.070**			-		
**Use of Face Towel**																		
No *	2168	142	6.55	444	26	5.86	404	21	5.20	665	67	10.08	258	11	4.26	397	17	4.28
Individual use	3563	217	6.09	694	52	7.49	616	26	4.22	536	53	9.89	900	60	6.67	817	26	3.18
Collective use	1684	87	5.17	393	23	5.85	300	16	5.33	452	28	6.19	203	6	2.96	336	14	4.17
OR ^c^ (95% CI)	0.92 (0.66–1.28)			1.47 (0.78–2.79)			0.78 (0.42–1.47)			0.98 (0.49–1.94)			2.01 (0.38–10.42)			0.84 (0.12–5.70)		
p-value	0.628			0.230			0.456			0.969			0.405			0.863		
OR ^c^ (95% CI)	0.72 (0.48–1.08)			1.04 (0.49–2.18)			1.02 (0.49–2.10)			0.49 (0.22–1.07)			0.56 (0.04–6.71)			1.05 (0.11–9.96)		
p-value	**0.122**			0.912			0.953			**0.074**			0.649			0.964		
**Use of Soap to Wash Face and Hands**																		
Always *	3395	164	4.83	779	50	6.42	511	28	5.48	871	49	5.63	534	13	2.43	700	24	3.43
Sometimes	2553	179	7.01	526	37	7.03	592	24	4.05	461	62	13.4	462	34	7.36	512	22	4.30
Never	1407	99	7..04	215	13	6.05	185	9	4.86	311	36	11.6	364	30	8.24	332	11	3.31
OR ^c^ (95% CI)	1.77 (1.28–2.44)			1.25 (0.70–2.23)			0.71 (0.39–1.31)			3.56 (1.86–6.79)			5.55 (1.74–17.7)			2.03 (0.65–6.37)		
p-value	**0.001**			0.453			0.278			**<0.001**			**0.004**			0.222		
OR ^c^ (95% CI)	1.67 (1.13–2.47)			0.91 (0.40–2.10)			0.87 (0.37–2.00)			2.74 (1.31–5.74)			6.62 (1.94–22.6)			1.07 (0.40–2.89)		
p-value	**0.010**			0.830			0.741			**0.008**			**0.003**			0.885		
**Sleeps Alone**																		
Yes *	2894	181	6.25	657	48	7.31	597	25	4.19	633	53	8.37	457	34	7.44	550	21	3.82
No	4529	265	5.85	878	53	6.04	725	38	5.24	1022	95	9.30	904	43	4.76	1000	36	3.60
OR ^c^ (95% CI)	0.92 (0.69–1.21)			0.80 (0.47–1.35)			1.27 (0.73–2.23)			1.18 (0.67–2.06)			0.64 (0.21–1.95)			0.71 (0.17–2.94)		
p-value	0.569			0.412			0.389			0.552			0.442			0.642		
**Presence of Nasal Secretion**																		
No *	6895	400	5.80	1405	90	6.41	1233	54	4.38	1514	136	8.98	1288	69	5.36	1455	51	3.51
Yes	528	46	8.71	130	11	8.46	89	9	10.11	141	12	8.51	73	8	10.96	95	6	6.32
OR ^c^ (95% CI)	1.96 (1.21–3.17)			1.55 (0.67–3.57)			2.67 (1.13–6.33)			1.05 (0.41–2.69)			0.50 (0.02–11.77)			6.97 (0.78–61.63)		
p-value	**0.006**			0.295			**0.025**			0.908			0.670			**0.081**		

a: Prevalence of trachoma; b: Percentages were calculated excluding missing values; c: Odds ratio corrected using random effect; * Reference.

**Table 2 pathogens-08-00263-t002:** Univariate analysis on variables relating to household characteristics of cases of trachoma in the population between 1 and 9 years of age investigated in Pernambuco and mesoregions, 2014–2015.

Household Characteristics	Pernambuco			Metropolitan Region of Recife			Zona da Mata			Agreste			Sertão do São Francisco			Sertão Pernambucano		
	N	+	% ^a,b^	n	+	% ^a,b^	n	+	% ^a,b^	n	+	% ^a,b^	n	+	% ^a,b^	n	+	% ^a,b^
**Type of home**																		
Masonry *	6816	411	6.03	1465	95	6.48	1.238	61	4.93	1539	136	8.84	1144	64	5.59	1430	55	3.85
Others	607	35	5.77	70	6	8.57	84	2	2.38	116	12	10.34	217	13	5.99	120	2	1.67
OR ^c^ (95% CI)	0.93 (0.55–1.57)			1.51 (0.52–4.34)			0.46(0.10–2.12)			1.20 (0.38–3.81)			1.28 (0.22–7.22)			0.43 (0.01–15.66)		
p-value	0.810			0.441			0.326			0.746			0.773			0.650		
**Water supplied from the public network**																		
Yes *	3048	155	5.09	898	56	6.24	686	32	4.66	223	13	5.83	654	33	5.05	587	21	3.58
No	4367	290	6.64	637	45	7.06	636	31	4.87	1431	135	9.43	705	44	6.24	958	35	3.65
OR ^c^ (95% CI)	1.45 (1.07–1.94)			1.16 (0.69–1.97)			1.04 (0.60–1.78)			1.96 (0.75–5.11)			1.28 (0.73–2.24)			1.10 (0.59–2.08)		
p-value	**0.015**			0.570			0.895			**0.167**			0.379			0.759		
**Intermittence of water supply**																		
No period without water *	5101	326	6.39	795	55	6.92	653	29	4.44	1509	138	9.15	946	59	6.24	1198	45	3.76
Periods without water	2297	117	5.09	735	46	6.26	667	32	4.80	141	10	7.09	403	17	4.22	351	12	3.42
OR ^c^ (95% CI)	0.75 (0.55–1.02)			0.87 (0.52–1.48)			1.09 (0.65–1.82)			0.67 (0.22–2.01)			0.84 (0.33–2.13)			1.01 (0.41–2.52)		
p-value	**0.074**			0.619			0.753			0.475			0.723			0.981		
**Type of sewage**																		
Public network/septic tank *	2220	113	5.09	575	32	5.57	563	23	4.09	348	25	7.18	354	21	5.93	380	12	3.16
Cesspit	2336	160	6.85	611	48	7.86	431	27	6.26	420	41	9.76	463	26	5.62	411	18	4.38
Other	2867	173	6.03	349	21	6.02	328	13	3.96	887	82	9.24	544	30	5.51	759	27	3.56
OR ^c^ (95% CI)	1.55 (1.08–2.22)			1.54 (0.85–2.78)			1.58 (0.85–2.92)			1.56 (0.64–3.77)			0.98 (0.20–4.69)			1.52 (0.15–15.14)		
p-value	**0.017**			**0.147**			**0.140**			0.321			0.986			0.719		
OR ^c^ (95% CI)	1.21 (0.84–1.72)			0.99 (0.46–2.10)			0.96 (0.47–1.98)			1.38 (0.62–3.09)			0.82 (0.16–4.09)			1.44 (0.18–11.20)		
p-value	0.303			0.985			0.933			0.423			0.816			0.723		
**Destination of sewage**																		
Toilet with flush *	3694	229	6.20	800	60	7.50	640	29	4.53	696	62	8.91	772	46	5.96	786	32	4.07
Other	3729	217	5.82	735	41	5.58	648	34	5.00	959	86	8.97	589	31	5.26	764	25	3.27
OR ^c^ (95% CI)	0.90 (0.67–1.20)			0.66 (0.39–1.11)			1.13 (0.65–1.97)			1.02 (0.56–1.85)			0.78 (0.41–1.47)			0.95 (0.53–1.70)		
p-value	0.472			**0.116**			0.656			0.960			0.437			0.866		
**Destination of solid waste**																		
Public collection *	3288	185	5.63	1334	85	6.37	766	36	4.70	393	35	8.91	288	16	5.56	507	13	2.56
Other forms	4133	261	6.32	201	16	7.96	556	27	4.86	1261	113	8.96	1073	61	5.68	1042	44	4.22
OR ^c^ (95% CI)	1.15 (0.87–1.54)			1.31 (0.62–2.78)			1.03 (0.59–1.78)			1.13 (0.56–2.28)			0.50 (0.15–1.70)			1.91 (0.81–4.51)		
p-value	0.324			0.483			0.921			0.732			0.270			**0.140**		
**Flies in the household**																		
No *	5511	343	6.22	553	35	6.33	281	8	2.85	277	30	10.83	161	5	3.11	619	24	3.88
Yes	1891	102	5.39	978	66	6.75	1037	54	5.21	1370	118	8.61	1197	72	6.02	929	33	3.55
OR ^c^ (95% CI)	1.20 (0.87–1.68)			1.01 (0.58–1.75)			1.94 (0.88–4.29)			0.66 (0.31–1.42)			2.09 (0.25–17.05)			1.00 (0.20–5.00)		
p-value	0.257			0.950			**0.100**			0.294			0.490			0.998		
**Number of rooms for sleeping**																		
One *	1074	50	4.66	318	15	4.72	163	9	5.52	190	16	8.42	185	4	2.16	218	6	2.75
Two	4175	261	6.25	895	71	7.93	764	34	4.45	885	80	9.04	749	45	6.01	882	31	3.51
Three or more	2033	129	6.35	291	15	5.15	365	17	4.66	546	49	8.97	415	28	6.75	416	20	4.81
OR ^c^ (95% CI)	1.60 (1.02–2.51)			2.14 (1.00–4.55)			0.80 (0.38–1.68)			1.28 (0.45–3.63)			3.23 (1.65–6.35)			1.27 (0.45–3.58)		
p-value	**0.042**			**0.047**			0.552			0.637			**0.001**			0.653		
OR ^c^ (95% CI)	1.67 (1.03–2.71)			1.18 (0.46–3.04)			0.84 (0.36–1.92)			1.42 (0.48–4.19)			4.73 (2.04–10.9)			1.69 (0.54–5.27)		
p-value	**0.036**			0.724			0.675			0.524			**<0.001**			0.363		
**Educational level of the head of the household**																		
9 years or more *	2294	130	5.67	552	41	7.43	403	19	4.71	272	27	9.93	533	25	4.69	534	18	3.37
Between 0 and 8 years	5102	309	6.06	975	58	5.95	917	44	4.80	1379	119	8.63	827	52	6.29	1004	36	3.59
OR ^c^ (95% CI)	1.06 (0.78–1.43)			0.76 (0.44–1.30)			1.00 (0.53–1.86)			0.79 (0.38–1.65)			1.16 (0.59–2.31)			1.07 (0.58–1.96)		
p-value	0.727			0.318			0.998			0.532			0.661			0.838		
**Family income (in minimum monthly wages, MW)**																		
More than 1 MW *	1038	58	5.59	415	23	5.54	193	8	4.15	99	7	7.07	156	9	5.77	175	11	6.29
Up to 1 MW	6355	387	6.09	1108	77	6.95	1128	55	4.88	1555	141	9.07	1193	68	5.70	1371	46	3.36
OR ^c^ (95% CI)	1.06 (0.71–1.59)			1.29 (0.71–3.36)			1.18 (0.52–2.63)			1.35 (0.39–4.61)			0.85 (0.36–2.02)			0.49 (0.22–1.11)		
p-value	0.764			0.395			0.690			0.634			0.717			0.089		
**Number of people in the household**																		
2 to 3 *	1369	71	5.19	275	22	8.00	242	12	4.96	277	16	5.78	247	14	5.67	328	7	2.13
4 to 5	3261	240	6.63	792	55	6.94	644	30	4.66	771	85	11.0	674	34	5.04	740	36	4.86
6 or more	2433	135	5.55	468	24	5.13	436	21	4.82	607	47	7.74	440	29	6.59	482	14	2.90
OR ^c^ (95% CI)	1.30 (0.93–1.81)			0.81 (0.42–1.54)			0.94 (0.45–1.93)			2.73 (1.13–6.64)			0.92 (0.47–1.82)			6.54 (1.29–33.1)		
p-value	**0.125**			0.520			0.861			**0.026**			0.820			**0.023**		
OR ^c^ (95% CI)	1.04 (0.72–1.50)			0.58 (0.27–1.22)			0.97 (0.44–2.16)			1.85 (0.73–4.69)			1.13 (0.54–2.37)			2.92 (0.61–14.1)		
p-value	0.830			**0.153**			0.953			**0.194**			0.738			**0.180**		

a: Prevalence of trachoma; b: Percentages were calculated excluding missing values; c: Odds ratio corrected using random effect adjusted by age; * Reference.

**Table 3 pathogens-08-00263-t003:** Final adjusted model for associations of individual and household characteristics of cases with trachoma in the population between 1 and 9 years of age investigated in Pernambuco and mesoregions, 2014–2015.

Characteristics	Pernambuco	Metropolitan Region of Recife	Zona da Mata	Agreste	Sertão do São Francisco	Sertão Pernambucano
OR (95% CI)	OR (95% CI)	OR (95% CI)	OR (95% CI)	OR (95% CI)	OR (95% CI)
**Sex**
Male	Reference		Reference			
Female	1.45 (1.10–1.90)		1.59 (0.93–2.72)			
p-value	0.008		0.093			
**Age group**
From 1 to 4	Reference					
From 5 to 9	1.34 (1.01–1.76)					
p-value	0.039					
**Use of soap to wash face and hands**
Always *	Reference			Reference	Reference	
Sometimes	1.77 (1.27–2.46)			4.03 (1.97–8.24)	2.43 (1.07–5.56)	
p-value	0.001			<0.001	0.035	
Never	1.61 (1.07–2.41)			3.11 (1.41–6.87)	2.49 (1.01–6.15)	
p-value	0.021			0.005	0.048	
**Presence of nasal secretion**
No *	Reference		Reference			
Yes	1.94 (1.15–3.27)		2.55 (1.13–5.79)			
p-value	0.013		0.025			
**Water supplied from public network**
Yes	Reference					
No	1.40 (1.03–1.91)					
p-value	0.033					
**Number of rooms for sleeping**
One *	Reference	Reference			Reference	
Two	1.66 (1.05–2.64)	2.14 (1.00–4.55)			3.16 (1.60–6.23)	
p-value	0.031	0.047			0.001	
Three or more	1.69 (1.04–2.77)	1.18 (0.46–3.04)			4.69 (1.97–11.2)	
p-value	0.036	0.724			<0.001	
**Family income (in minimum monthly wages, MW)**
More than 1 MW *					Reference
Up to 1 MW					0.50 (0.22–1.15)
p-value					0.100
Number of people in the household

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
