# Peer review of "Household Survey of Trachoma among Children Living in Pernambuco, Brazil"

_pathogens, 2019, doi:10.3390/pathogens8040263_

Round 1

Reviewer 1 Report

Abstract - Line 17: incidence or prevalence?

Figure 1 looks blurry. Authors should submit a figure with better resolution in the final files.

Line 66: population-based household study seems more appropriate to describe this study design

Reviewer 2 Report

The manuscript has improved during the revision process.

Author Response

This manuscript is a resubmission of an earlier submission. The following is a list of the peer review reports and author responses from that submission.

Round 1

Reviewer 1 Report

This cross-sectional study by Brito et al describes the prevalence of trachoma among individuals aged 1-9 years old living in Pernambuco state, Northeast Brazil. The study reveals a high prevalence of trachoma in this setting and highlights the need for continuing monitoring of this disease. Overall, the manuscript needs to be improved by clarifying the methods and lightening the writing. There are several typographical and punctuation mistakes along the text. In addition, poor structure makes some sentences difficult to interpret. I suggest having the entire manuscript proofread for English.

Introduction: Some sentences are too vague (e.g. line 33 “it’s transmitted directly”: eye to eye contact, hands??)

Methods: The structure of this section should be improved. Please consider divide this section into subheadings (suggestions: Study population, Sample size, Data collection, Case definition, Ethical statement, Statistics). Information is all over the place which makes it difficult for the reader to follow. Moreover, a more direct explanation of some terms used are needed.

Lines 69-74: Please re-write this sentence. It’s hard to understand. What do the authors mean by ‘enumeration areas’?

Figure 1: Please improve this figure. It’s quite hard to distinguish the five mesoregions based on the chosen shaded colors.

Results: Lines 132-134: Please add appropriate statistical test to support this statement.

Figure 2: Can you show that by mesoregions?

This manuscript would greatly benefit from a risk factors analysis. Considering that the data used in the study was derived from a secondary database, would it be possible to include other relevant variables (socio-economic, demographic and household characteristics)?

Reviewer 2 Report

The manuscript describes prevalence of trachoma in Brazilian households in Pernambuco area. Trachoma still exists and causes significant morbidity. 

Major comments:

I suggest that the language is edited, otherwise it is impossible to understand properly the text and evaluate the manuscript.

Are there any publications based on this material elsewhere/in Portuguese? (Abstract mentions something about secondary data from National Household Survey on Trachoma 2014-2015)

Figure 1 and Figure 2 contain partly same data

Did you analyse risk factors for trachoma in these settings? That would make the study stronger.